# An Automated Wavelet-Based Sleep Scoring Model Using EEG, EMG, and EOG Signals with More Than 8000 Subjects

**DOI:** 10.3390/ijerph19127176

**Published:** 2022-06-11

**Authors:** Manish Sharma, Anuj Yadav, Jainendra Tiwari, Murat Karabatak, Ozal Yildirim, U. Rajendra Acharya

**Affiliations:** 1Department of Electrical and Computer Science Engineering, Institute of Infrastructure, Technology, Research and Management (IITRAM), Ahmedabad 380026, India; anuj.yadav.17e@iitram.ac.in (A.Y.); jainendra.tiwari.17e@iitram.ac.in (J.T.); 2Department of Software Engineering, Firat University, Elazig 23119, Turkey; mkarabatak@firat.edu.tr (M.K.); ozalyildirim@firat.edu.tr (Ö.Y.); 3Department of Electronics and Computer Engineering, Ngee Ann Polytechnic, Singapore 599489, Singapore; aru@np.edu.sg; 4Department of Bioinformatics and Medical Engineering, Asia University, Taichung 41354, Taiwan; 5Department of Biomedical Engineering, School of Science and Technology, Singapore University of Social Sciences, Singapore 599494, Singapore

**Keywords:** polysomnogram (PSG), ensemble bagged tree (EBT), sleep stages, EEG, EMG, EOG, Tsallis entropy, Cohen’s kappa coefficient, wavelet decomposition

## Abstract

Human life necessitates high-quality sleep. However, humans suffer from a lower quality of life because of sleep disorders. The identification of sleep stages is necessary to predict the quality of sleep. Manual sleep-stage scoring is frequently conducted through sleep experts’ visually evaluations of a patient’s neurophysiological data, gathered in sleep laboratories. Manually scoring sleep is a tough, time-intensive, tiresome, and highly subjective activity. Hence, the need of creating automatic sleep-stage classification has risen due to the limitations imposed by manual sleep-stage scoring methods. In this study, a novel machine learning model is developed using dual-channel unipolar electroencephalogram (EEG), chin electromyogram (EMG), and dual-channel electrooculgram (EOG) signals. Using an optimum orthogonal filter bank, sub-bands are obtained by decomposing 30 s epochs of signals. Tsallis entropies are then calculated from the coefficients of these sub-bands. Then, these features are fed an ensemble bagged tree (EBT) classifier for automated sleep classification. We developed our automated sleep classification model using the Sleep Heart Health Study (SHHS) database, which contains two parts, SHHS-1 and SHHS-2, containing more than 8455 subjects with more than 75,000 h of recordings. The proposed model separated three classes if sleep: rapid eye movement (REM), non-REM, and wake, with a classification accuracy of 90.70% and 91.80% using the SHHS-1 and SHHS-2 datasets, respectively. For the five-class problem, the model produces a classification accuracy of 84.3% and 86.3%, corresponding to the SHHS-1 and SHHS-2 databases, respectively, to classify wake, N1, N2, N3, and REM sleep stages. The model acquired Cohen’s kappa (κ) coefficients as 0.838 with SHHS-1 and 0.86 with SHHS-2 for the three-class classification problem. Similarly, the model achieved Cohen’s κ of 0.7746 for SHHS-1 and 0.8007 for SHHS-2 in five-class classification tasks. The model proposed in this study has achieved better performance than the best existing methods. Moreover, the model that has been proposed has been developed to classify sleep stages for both good sleepers as well as patients suffering from sleep disorders. Thus, the proposed wavelet Tsallis entropy-based model is robust and accurate and may help clinicians to comprehend and interpret sleep stages efficiently.

## 1. Introduction

Sleep is a daily routine in human life. Sleep can be categorized into two phases, namely, rapid eye movement (REM) sleep, and non-REM (NREM) sleep, as per the Rechtschaffen and Kales (R&K) guidelines [1]. A healthy adult’s one-night sleep contains 4 to 5 cycles of REM and NREM sleep; each cycle lasts for approximately 90 min. NREM sleep is further categorized into four phases: S1, S2, S3, and S4. Sleep stages S3 and S4 are combined into a stage called N3, as per the American Academy of Sleep Medicine’s (AASM) [2] guidelines. Thus, according to the AASM, the NREM stage consists of only three sleep stages, namely N1, N2, and N3 [3,4].

Proper sleep-stage scoring helps doctors diagnose sleep disorders and decide on the relevant course of action for the treatment. The present gold standard for sleep analysis is the scoring of visual sleep phases performed manually by human experts. Sleep stages are graded by technicians and clinicians associated with a visual inspection of neurophysiologic signal patterns [5]. A hypnogram is a visual description of sleep phases throughout the night. It provides a simple picture of sleep that can be used to suspect or confirm sleep problems [6]. Analyzing the hypnograms over a night is a time-consuming process involving the subjectivity of sleep scorers. When given the same polysomnography (PSG) recording, different human sleep-scoring specialists are likely to develop varied sleep-staging assessments, and agreement among them may be around 83% [7]. The sleep-scoring accuracy of individual human experts is about 80% [8]. Even though the same specialist analyzes the same recording twice, the results may be slightly different, and the agreement between both sleep-scoring assessments would be, at most, 90% [9]. Sleep stages help as an asymptomatic instrument to identify sleep issues. The sleep scoring is often performed in a clinical setting by professionals using PSG [10] monitoring, which commonly includes electroencephalogram (EEG), electrooculogram (EOG), electrocardiogram (ECG) data, and electromyogram (EMG) signals, as per the R&K [11] principles or the AASM guidelines [2,12,13]. PSG recordings are first segmented into 30 s epochs, complying with the AASM guidelines. PSG is an effective tool used in sleep medicine to record many biological signals to determine sleep quality. When the patient sleeps, multiple signals, such as overnight EEG, breathing through the nostrils, breathing rate, blood pressure variations, ECG signals, blood oxygen content, EOG signals, and contact EMGs on the chin and legs, are recorded [14]. However, overnight recording using multiple channels causes discomfort in patients, and such data sometimes may not represent natural sleep patterns. PSG analysis for sleep-stage scoring is a costly affair requiring dedicated sleep labs and expensive equipment [15,16,17,18,19]. Recording sleep in a sleep lab cannot be conducted for multiple nights, and only one or two nights’ worth of sleep data may not accurately represent the complete scenario. For accurate sleep analysis, sleep recordings of multiple nights are required, which demands a portable and home-based system that captures only a few desired signals and requires the placement of only a few electrodes and sensors on the subject’s body. The proposed study explores the effectiveness of various channels and their combination for three signals, namely, EEG, EOG, and EMG, which are considered to be the most essential for sleep analysis. Further, this study aims to obtain a simple system that uses a minimum number of electrodes, but at the same time has a comparable performance with PSG-based systems such that it can be used as a portable home-based sleep-scoring system [12,20,21].

The National Sleep Research Resource (NSSR) contains PSG data of the SHHS [22,23] database, which was used for this study. The SHHS is a polycentric cohort developed to observe sleep-disordered ventilation related to an increased chance of heart disease [6]. The PSG recordings of 5793 subjects were collected during the 1995–1998 period. This database is known as the SHHS-1. For the remaining 2651 subjects, recordings were collected during the 2001–2003 period; these are known as the SHHS-2 database. The sampling frequency of both the EEG and EMG signals is 125 Hz, whereas the sampling frequency of the EOG signal is 50 Hz. PSGs were annotated by certified expert sleep scorers in both sessions using modified R&K standards. A single clinician manually assessed every record for sleep stages in 30-s epochs using R&K [11] scoring standards, obtaining multiple sleep stages. The details of both databases are given in Table 1.

It is to be noted that few studies have been performed in the literature to identify sleep stages using the SHHS-1 database. However, to the best of our knowledge, no study has used both SHHS-1 and SHHS-2. Sors et al. [6] employed a convolutional neural network using only the SHHS-1 database to score sleep stages using EEG and obtained an accuracy of 87% and a Cohen’s κ coefficient of 0.81. Biswal et al. [24] conducted a study on sleep-stage classification employing recurrent and convolutional neural networks (RCNN) and achieved 77.9% accuracy for the SHHS-1 dataset. Zhang et al. [5] also used the SHHS-1 dataset for sleep scoring using spectrograms from raw data. In addition, they used recurrent and convolutional neural networks to score sleep stages and obtained 87% accuracy. Zhang et al. [25] performed sleep-stage classification using a bidirectional long short-term memory (BLSTM) neural network to assess the four sleep stages (deep sleep, light sleep, wake, REM) and obtained a prediction accuracy of 80.25%. Fernández-Varela et al. [26] used only 500 random sleep recordings of the SHHS-1 database and achieved an accuracy of 75% by using a convolutional network. Wongsirichot et al. [27] used the k-nearest neighbors classifier with 14 bio-medical channels of the SHHS-2 database and found an accuracy of 83.76%.

In this work, we introduce an automatic sleep-stage scoring system using a wavelet-based Tsallis entropy feature, wherein the model is developed employing the SHHS-1 and SHHS-2 databases together. The deployment of both datasets makes the model robust. The SHHS-1 and SHHS-2 databases contain 5804 and 2651 subjects, respectively. We used C3-A2 and C4-A1 channels for the EEG signal—one EMG channel and two EOG channels (EOG-L and EOG-R). Thus, in contrast to earlier studies that used SHHS-1 only, the proposed study employs a comprehensive database comprising both SHHS-1 and SHHS-2. The wavelet decomposition of the signals is also performed using a new optimal orthogonal filter bank. Furthermore, we used individual channels as well as their varying combinations. We have explored the performance of 15 varying combinations. This is the first study on a joint SHHS database that used the optimal wavelet-based single feature to score sleep stages.

## 2. Methodology

Figure 1 shows the flow diagram for the suggested system, representing the actions taken in scoring sleep stages. Data acquisition, segmentation, wavelet decomposition, feature extraction, and classification are all part of the process. The PSG data were collected and segmented into an epoch of 30 s. The six sub-bands were acquired using a five-level wavelet decomposition using an orthogonal wavelet filter bank. The Tsallis entropy feature is extracted for these sub-bands. The features are then fed into machine learning classifiers to identify sleep stages.

### 2.1. Data Gathering

The SHHS database contains 8444 PSG recordings, of which 8326 (5791 from visit-1 and 2535 from visit-2) were selected for this study based on the availability of target signals. We used two bipolar EEG channels, two EOG channels, as well as an EMG channel. Every PSG recording was retrieved as a European data format (.edf) file, which also includes XML files for each subject. Every sleep stage of a 30-s epoch was annotated in the XML file, according to the R&K [11] rules. Recording files, the frequency of the channels, and the header details were obtained from the .edf file. The recording file contains the signal data in matrix form, which is normalized for segmentation into an epoch of 30 s. A total of 5,861,304 and 3,037,838 epochs were obtained from the SHHS-1 and SHHS-2 databases. The details of the sleep-stage epoch distribution for both visits are given in Table 1.

### 2.2. Orthogonal Wavelet Filter Bank

Instead of using Daubechies standard wavelet dB filters [28,29], we used an optimal orthogonal wavelet filter in this study [30]. The orthogonal filter employed has a minimum time-frequency product [31]. For the chosen length, first, an optimal half-band filter was designed, formulating a convex optimization problem in the form semidefinite program such that the spectral factors had minimum mean squared bandwidths. Then, the optimal spectral factor possessing a minimum mean squared duration was selected. Thus, the low-pass filter chosen had minimum spread both in frequency and time [31,32]. We used a filter with three vanishing moments and with a length of 18 in this study.

### 2.3. Wavelet Decomposition

The wavelet decomposition of level five was applied to every epoch of the PSG signals using the orthogonal wavelet filter bank. As a result, we obtained six sub-bands, out of which one was an approximate band, and the remaining five were detailed bands.

### 2.4. Feature Extraction

Feature extraction was performed by extracting the Tsallis entropy features from each sub-band to classify sleep stages. Tsallis entropy is important in non-extensive statistics because it accurately explains complicated systems’ statistical properties. Tsallis entropy is regarded as a valuable metric for defining the thermo-statistical features of a specific class of given system, which includes long-range connections, long-term memories, and multi-fractal systems [33]. The generalization of Boltzmann–Gibbs–Shannon (BGS) entropy [34] gives Tsallis entropy (TE):$$TE=\sum_{i=1}^{n}x_{i}-{x_{i}}^{2}$$
where $x_{i}$ denotes the $i_{th}$ sample of the wavelet coefficient sequence *x*(*n*) of length N.

### 2.5. Classification Method

Wavelet-based Tsallis entropy features are used to classify sleep stages. Two classification tasks were considered: (i) 3-class classification for distinguishing W vs. NREM vs. REM, and (ii) a 5-class task to discriminate 5 classes, namely, W vs. REM vs. N1 vs. N2 vs. N3. The extracted labeled features were fed to the machine learning classifier, namely, an ensemble bagged decision tree with 10% holdout validation [31,35]. Ensemble learning is a cutting-edge method to solve various machine learning issues by integrating the outputs of many base learners [36,37]. A bagging method and decision tree classifier were combined in the EBT classifier. Bagging is a strategy that uses bootstrap sampling to decrease decision tree variance and increase learning algorithm efficiency by producing a group of learning algorithms that are trained in succession. It employs arbitrary sampling with a replacement rather than a conventional averaging of all outcomes from multiple decision trees [38].

## 3. Results

The present approach is conducted on an Intel^®^ Core™ i5 eighth-generation CPU @ 1.60 GHz, 8 GB RAM, and Windows 10 (64-bit) OS with MATLAB R2016a. For the classification task, the proposed method employs 8,899,142 epochs (5,861,304 epochs for visit-1 and 3,037,838 epochs for visit-2). A summary of the results obtained for automated sleep-stage scoring with the ensemble bagged tree classifier using 10% holdout validation to classify sleep stages into three stages using the SHHS (1 and 2) datasets can be observed in Table 2 and Table 3, respectively. Additionally, Table 4 and Table 5 summarize the categorization results for five sleep stages.

Signals are divided into six sub-bands using a five-level one-dimensional decomposition. Tsallis entropy is then used to extract features, which are then loaded into the EBT classifier for three-stage and five-stage classification. We merged the sleep stages N1, N2, and N3, and represented them as N for three sleep-stage classifications in this work. Table 2 and Table 3 exhibit each channel’s accuracy alone and collectively for three-class sleep-stage categorization, as acquired from visit-1 and visit-2. It also shows that the maximum accuracy is yielded by integrating all channels rather than individual channels. For three-stage classification using visit-1, by combining all the channels, we scored an accuracy of 90.70% and a Cohen’s κ coefficient of 0.833, whereas, for visit-2, we achieved 91.80% accuracy and a Cohen’s κ coefficient of 0.86. The confusion matrix [39,40] for the three-class sleep-stage classification for both datasets is shown in Table 6. Similarly, for the five-stage classification, by employing all the channels using visit-1, we achieved 84.30% classification accuracy with a Cohen’s κ value of 0.774, and 86.30% accuracy with Cohen’s κ coefficient of 0.80 for visit-2. The confusion matrix for the five-class sleep-stage classification is shown in Table 7.

## 4. Discussion

The PSG is widely recognized as a key element for assessing sleep phases and diagnosing sleep disorders. PSG-based approaches require the use of many connected sensing devices to capture the actions of various biomedical signals, as well as time-consuming analytic methods. Furthermore, the sleep measurements must be performed nightly in a professional sleep lab or clinic. As a result, it is essential to investigate novel approaches that can yield reliable outputs that are comparable to conventional sleep-staging PSG-based techniques, but that are simpler, less costly, and more comfortable for patients. The suggested method addresses all of the issues associated with earlier methods by employing a basic and straightforward methodology to classify sleep stages reliably.

This proposed study has provided an automatic sleep-stage classification system employing PSG signals. The suggested study considers a five-class and a three-class sleep-stage classification method. We used two unipolar EEG channels (C4-A1 and C3-A2), one EMG channel, and two EOG channels (EOG-L and EOG-R) separately and in combination for the present study. In addition, we used an orthogonal filter bank to decompose PSG epochs using a five-level one-dimensional wavelet filter bank.

As shown in Table 8, the proposed technique has obtained the highest classification accuracies compared to prior studies for three-class and five-class problems. The proposed approach acquired 90.70% and 91.80% accuracy for the three-stage classification by using the visit-1 and visit-2 datasets, respectively. Morevoer, the technique obtained a higher Cohen’s κ coefficient for both datasets compared to previous studies. Biswal et al. [24] employed only the SHHS-1 dataset using RCNN to categorize five-class sleep stages and achieved an accuracy of 77.90% while using only two uni-polar channels. The current study uses both datasets (visit-1 and visit-2) and EEG, EMG, and EOG channels both separately and together. Furthermore, the proposed study focuses on three- and five-stage categorization and achieves high accuracy. Sors et al. and Seo et al. [6,41] employed a single dataset (visit-1) and a single EEG channel to categorize five sleep phases while the suggested method employs a large dataset (visit-1 and visit-2) to score sleep stages. Linda Zhang et al. and Fernandez-Varela et al. [5,26] used a single dataset with three channels (EEG, EMG, and EOG) for five-class classification. Additionally, Wongsirichot et al. [27] used the visit-2 dataset with 14 biomedical channels for 5-class stage classification using a k-nearest neighbor classifier, and achieved an accuracy of 83.76%, which is less than the proposed method. The proposed technique used five channels for three-stage and five-stage classification and employs an ensemble bagged tree classifier, and achieved higher classification accuracy.

The current work is unique because it gives an idea about how different PSG signals contribute to the sleep-scoring capability of a machine learning model. We observed that EEG signals are the most effective in sleep scoring, followed by EOG and EMG signals. A combination of these channels yields even better sleep-scoring capabilities, and helps develop a system that can score sleep stages in an effective, rapid, and simple manner. The following are some additional advantages of the proposed study:This is the first study that employs a huge database with two subsets, visit-1 with 5791 subjects and visit-2 with 2535 subjects, and it is also the first study to employ more epochs (5,861,304 for visit-1 and 3,037,838 for visit-2) than previous research [5,6,24,41]. We have used 75468 h of sleep recordings (48861 h in visit-1 and 26607 h in visit-2);The proposed model demonstrated high recognition accuracy for the three- and five-stage classification tasks examined in this study. Table 2, Table 3, Table 4 and Table 5 show that the EMG is the least-accurate signal in sleep-scoring tasks, whereas EEG is the most-accurate one. EOG-R and EOG-L had similar scoring capabilities, and taking only one did not significantly change the outcome. A combination of all five target signals yielded the best-performing model;We have used orthogonal filters to create a new five-level, one-dimensional wavelet decomposition class. Tsallis entropy is used to extract the features from every sub-band;We extracted Tsallis entropy-based features from two EOG channels, two uni-polar EEG channels, and one EMG, with sampling frequencies of 125 Hz, 50 Hz, and 125 Hz. This made the process easier and more computationally efficient;A system for detecting sleep stages that is simple, rapid, and accurate is being developed;For both SHHS sleep datasets, the suggested model produced high Cohen’s κ coefficient values (greater than 0.75).

The limitations of the proposed method are as follows:Because of the restricted amount of data available in an unbalanced database, the proposed method achieves lower accuracy in categorizing the N1 sleep stage (SHHS). However, balancing the employed data improves the classification of the N1 sleep stage;Wavelet-based features take longer to compute than traditional statistical features. However, interestingly, the same wavelet filter is used to remove noise;The suggested method employs PSG signals with many channels, which may cause a little inconvenience to individuals. The portable recording machines that can be used at homes are also more expensive;Since the database has many participants, the machine learning classifier takes longer to categorize sleep stages.

Deep learning (DL) is now widely used to categorize biomedical signals [42]. However, DL-based algorithms perform well when dealing with large databases [43]. As a result, we plan to incorporate deep learning (DL) techniques such as recurrent neural networks (RNN), long short-term memory (LSTM), and auto-encoders into our future study.

## 5. Conclusions

The identification of the sleep stages plays a significant role in sleep science. Normal sleep rating in a sleep clinic utilizing human PSG records is costly and difficult for experts. In certain ways, defining the sleep cycle process will reduce costs and accelerate sleep research. In this work, we propose a technique for automatic sleep-stage classification using PSG signals. We employed two unipolar EEG channels (C4-A1 and C3-A2), one EMG channel, and two EOG channels (EOG-L and EOG-R) individually and in various combinations for three-class and five-class classification tasks. We used the PSG signal recordings of 5791 subjects from the visit-1 dataset, and 2535 subjects from the visit-2 dataset. The PSG signals were segmented into numerous 30 s epochs corresponding to three and five classes, and each epoch was subjected to a five-level, one-dimensional wavelet decomposition using an orthogonal filter bank. This was followed by computing the Tsallis entropy-based features for each sub-band.

The proposed method employs an EBT classifier with 10% holdout validation for the three-class and five-class classification problems. The proposed method achieved a maximum accuracy of 90.70% with the visit-1 dataset and of 91.80% with the visit-2 dataset for three-class classification, while for the five-class sleep-stage classification, we achieved an accuracy of 84.30% with visit-1 and 86.30% with visit-2. Further, when the visit-1 and visit-2 datasets were used, the model produced higher Cohen’s κ coefficients (0.838 (visit-1) and 0.86 (visit-2) for the three-class classification and 0.7746 (visit-1) and 0.8007(visit-2) for the five-class one, respectively). The proposed model’s classification accuracy reveals that it can effectively categorize sleep stages utilizing 30 s duration and PSG signals, and it can be employed in home-based systems and clinics to classify sleep stages.

## Figures and Tables

**Figure 1 ijerph-19-07176-f001:**
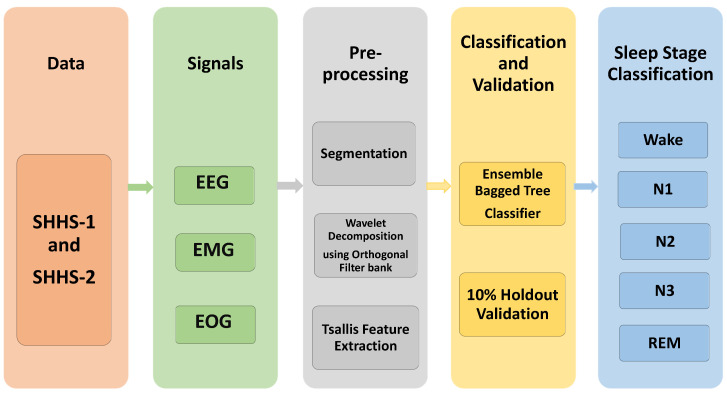
Block diagram of the proposed work.

**Table 1 ijerph-19-07176-t001:** Summary of the SHHS database.

Variable	SHHS-1				SHHS-2			
Subject Count	5793				2651			
Size of Dataset	216 GB				137 GB			
Male Patients	3033				1425			
Female Patients	2760				1226			
	mean	std	min	max	mean	std	min	max
Age (year)	63.14	11.23	39	90	67.23	10.38	44	90
Body Mass Index (BMI)	28.16	5.09	18	50	28.31	5.05	18	50
Epworth Sleepiness Scale (ESS) score	7.77	4.4	0	24	7.51	4.21	0	24
Total Sleep Time (minutes)	506.07	37.36	180	599.5	602.15	68.52	261	845.5
Wake (%)	28.72	12.29	1.56	91.42	37.43	11.62	7.29	88.16
Repid Eye Moment (REM) Sleep (%)	13.96	5.75	0	35.73	12.97	5.15	0	34.31
Non-REM stage 1 (%)	3.7	2.62	0	23.8	3.51	2.9	0	76.35
Non-REM stage 2 (%)	40.98	11.43	3.69	93.64	36.18	9.46	0	83.43
Non-REM stage 3 (%)	11.84	7.97	0	53.84	9.49	6.89	0	43.82
Sleep Efficiency (%)	71.28	12.29	8.58	98.44	62.57	11.62	11.84	92.71
Total Epochs	5,861,304				3,037,838			

**Table 2 ijerph-19-07176-t002:** Classification performance using different combinations of channels for classifying three sleep stages using SHHS-1.

Signals	Accuracy(%)				Cohen’s
	W	NREM	REM	Overall	Kappa
EMG	83.06	72.28	87.46	71.60	0.4552
EOG-R	86.26	81.78	87.44	77.75	0.5941
EOG-L	86.41	81.75	87.36	77.75	0.5941
EEG (C4-A1)	91.36	84.51	88.78	82.00	0.8305
EEG (C3-A2)	91.29	84.34	88.42	82.35	0.6839
EEG (C4-A1 + C3-A2)	93.22	87.63	90.99	86.00	0.7489
EMG + EOG-R	89.87	85.01	91.66	83.25	0.6975
EMG + EOG-L	89.93	85.01	91.62	83.25	0.6977
EOG-R + EOG-L	88.52	84.85	88.82	81.15	0.6581
EMG + EOG-R + EOG-L	90.46	86.23	92.15	84.40	0.7188
C3-A2 + C4-A1 + EMG	94.08	89.77	93.44	88.65	0.798
C3-A2 + C4-A1 + EOG-R + EOG-L	94.60	90.87	93.41	89.40	0.8109
C3-A2 + C4-A1 +EMG +EOG-R	95.00	91.67	94.46	90.55	0.8316
C3-A2 + C4-A1 +EMG +EOG-L	94.98	91.62	94.41	90.50	0.8109
**C3-A2 + C4-A1 + EMG + EOG-R + EOG-L**	**95.05**	**91.81**	**94.52**	**90.70**	**0.8338**

**Table 3 ijerph-19-07176-t003:** Classification performance using different combinations of channels for classifying three sleep stages using SHHS-2.

Signals	Accuracy(%)				Cohen’s
	W	NREM	REM	Overall	Kappa
EMG	83.05	72.92	88.09	72.00	0.5073
EOG-R	87.40	83.90	88.39	79.80	0.6533
EOG-L	87.33	83.84	88.44	79.80	0.6526
EEG (C4-A1)	91.90	86.01	89.49	83.20	0.7122
EEG (C3-A2)	91.61	85.73	89.02	83.70	0.7217
EEG (C4-A1 + C3-A2)	93.60	88.74	91.45	86.90	0.7766
EMG + EOG-R	90.87	86.82	92.52	85.10	0.7456
EMG + EOG-L	90.85	86.88	92.59	85.10	0.7461
EOG-R + EOG-L	88.73	86.21	90.04	82.50	0.6994
EMG + EOG-R + EOG-L	91.45	88.18	93.20	86.40	0.7683
C3-A2 + C4-A1 + EMG	94.37	90.79	94.05	89.60	0.8237
C3-A2 + C4-A1 + EOG-R + EOG-L	95.14	92.15	93.97	90.60	0.8406
C3-A2 + C4-A1 + EMG + EOG-R	95.43	92.78	95.05	91.60	0.8578
C3-A2 + C4-A1 + EMG + EOG-L	95.34	92.67	95.04	91.50	0.8561
**C3-A2 + C4-A1 + EMG + EOG-R + EOG-L**	**95.44**	**92.92**	**95.14**	**91.80**	**0.8600**

**Table 4 ijerph-19-07176-t004:** Classification performance using different combinations of channels for classifying five sleep stages using SHHS-1.

Signals	Accuracy(%)						Cohen’s
	W	N1	N2	N3	REM	Overall	Kappa
EMG	83.39	96.36	65.90	88.52	86.93	59.25	0.4125
EOG-R	87.75	96.60	80.05	92.05	88.88	67.95	0.5733
EOG-L	85.67	95.95	76.62	90.63	86.80	67.85	0.5328
EEG(C4-A1)	91.13	96.01	79.69	93.18	88.57	74.30	0.6303
EEG(C3-A2)	91.08	96.02	79.92	93.36	88.21	74.30	0.6301
EEG(C4-A1 + C3-A2)	92.83	96.18	83.48	94.64	90.87	79.10	0.6970
EMG + EOG-R	89.45	96.16	80.13	92.44	91.50	74.85	0.6354
EMG + EOG-L	89.47	96.18	80.15	92.47	91.35	74.80	0.6350
EOG-R + EOG-L	86.99	95.94	78.86	91.62	88.47	70.90	0.5787
EMG + EOG-R + EOG-L	90.06	96.20	81.61	93.03	92.14	76.50	0.6600
C3-A2 + C4-A1 + EMG	93.84	96.26	85.88	95.31	93.35	82.30	0.7462
C3-A2 + C4-A1 + EOG-R + EOG-L	94.45	96.19	86.81	95.20	93.28	83.00	0.7553
C3-A2 + C4-A1 + EMG + EOG-R	94.73	96.25	87.37	95.53	94.37	84.15	0.7720
C3-A2 + C4-A1 + EMG + EOG-L	94.76	96.31	87.30	95.51	94.28	84.05	0.7714
**C3-A2 + C4-A1 + EMG + EOG-R + EOG-L**	**94.79**	**96.25**	**87.55**	**95.57**	**94.45**	**84.30**	**0.7746**

**Table 5 ijerph-19-07176-t005:** Classification performance using different combinations of channels for classifying five sleep stages using SHHS-2.

Signals	Accuracy(%)						Cohen’s
	W	N1	N2	N3	REM	Overall	Kappa
EMG	81.64	96.31	66.67	89.99	86.99	60.80	0.4119
EOG-R	86.95	96.24	79.91	92.84	87.92	71.90	0.5864
EOG-L	86.88	96.29	80.00	92.84	88.03	72.00	0.5878
EEG(C4-A1)	91.76	96.25	82.40	94.71	89.26	76.70	0.6605
EEG(C3-A2)	91.31	96.25	82.33	95.09	88.70	77.20	0.6657
EEG(C4-A1 + C3-A2)	93.43	96.41	85.92	96.05	91.29	81.50	0.7301
EMG + EOG-R	90.40	96.40	82.90	94.14	92.35	78.10	0.6788
EMG + EOG-L	90.50	96.38	83.10	94.18	92.43	78.30	0.6812
EOG-R + EOG-L	88.17	96.30	82.13	93.61	89.62	74.90	0.6308
EMG + EOG-R + EOG-L	91.02	96.40	84.26	94.54	93.11	79.70	0.7016
C3-A2 + C4-A1 + EMG	94.14	96.45	87.58	96.32	93.97	84.20	0.7697
C3-A2 + C4-A1 + EOG-R + EOG-L	94.93	96.41	88.90	96.41	93.81	85.20	0.7844
C3-A2 + C4-A1 + EMG + EOG-R	95.19	96.43	89.20	96.51	94.96	86.10	0.7979
C3-A2 + C4-A1 + EMG + EOG-L	95.18	96.43	89.15	96.49	94.98	86.10	0.7975
**C3-A2 + C4-A1 + EMG + EOG-R + EOG-L**	**95.20**	**96.44**	**89.35**	**96.55**	**95.13**	**86.30**	**0.8007**

**Table 6 ijerph-19-07176-t006:** Confusion matrix relating to three sleep stages’ classification by using combined signals with 10% hold-out validation.

		Predicted class					Predicted class		
		Wake	N	REM			Wake	N	REM
True class	Wake	92%	7%	1%	True class	Wake	95%	5%	1%
	N	3%	95%	2%		N	3%	95%	2%
	REM	6%	23%	71%		REM	7%	21%	72%
SHHS-1					SHHS-2				

**Table 7 ijerph-19-07176-t007:** Confusion matrix relating to five sleep stages’ classification by using combined signals with 10% hold-out validation.

		Predicted class							Predicted class				
		Wake	N1	N2	N3	REM			Wake	N1	N2	N3	REM
True class	Wake	93%	1%	4%	1%	1%	True class	Wake	96%	<1%	3%	<1%	1%
	N1	24%	12%	47%	<1%	17%		N1	26%	11%	50%		14%
	N2	3%	<1%	90%	4%	3%		N2	3%	1%	90%	3%	2%
	N3	1%	<1%	22%	77%	<1%		N3	<1%		22%	77%	<1%
	REM	6%	1%	17%	<1%	75%		REM	7%	1%	16%	<1%	75%
SHHS-1							SHHS-2						

**Table 8 ijerph-19-07176-t008:** Comparison with existing state-of-the-art approaches in terms of accuracy and Cohen’s κ.

Study	Database	Subject	Signal	Accuracy		Cohen’s κ	
				C = 3	C = 5	C = 3	C = 5
Sors et al. [6]	SHHS-1	5793	C4-A1	-	87%	-	0.81
Biswal et al. [24]	SHHS-1	5791	C4-A1, C3-A2	-	77.90%	-	0.73
Linda zhang et al. [5]	SHHS-1	5793	EEG + EMG + EOG	-	87%	-	0.82
Fernandez-Varela et al. [26]	SHHS-1	500	EEG + EMG + EOG	-	78%	-	0.83
Wongsirichot et al. [27]	SHHS-2	2535	14 Biomedical	-	83.70%	-	N/A
Seo et al. [41]	SHHS-1	5791	C4-A1	-	86.30%	-	0.81
**Proposed Work**	**SHHS-1**	**5791**	**EEG + EMG + EOG**	**90.70%**	**84.30%**	**0.83**	**0.77**
	**SHHS-2**	**2535**	**EEG + EMG + EOG**	**91.80%**	**86.30%**	**0.86**	**0.80**

## Data Availability

Data used in this work is available upon request on National Sleep Research Resource.

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
