# Peer review of "An Automated Wavelet-Based Sleep Scoring Model Using EEG, EMG, and EOG Signals with More Than 8000 Subjects"

_ijerph, 2022, doi:10.3390/ijerph19127176_

Round 1
Reviewer 1 Report
The article is devoted to a topical and important topic - the construction of an automated sleep score system. Overall, the article is well written and makes a serious impression. There are a few comments on minor corrections.
line 39: the abbreviation is used, but it is introduced later in the text - in line 44.
It is better to describe the datasets earlier in the text. I suggest doing it in line 61.
line 90: it is necessary to clarify who annotated.
In the first paragraph from Material, it would be good to describe the results of the right side of Table 2.
The datasets use data on people of a certain age. The authors need to describe what limitations this places on the developed solution.
Author Response
Reviewer #1
Comment: The article is devoted to a topical and important topic - the construction of an automated sleep score system. Overall, the article is well written and makes a serious impression. There are a few comments on minor corrections.
Response: Thank you so very much for your kind comments and appreciation. We have made the suggested minor corrections in the revised version of the manuscript.
Comment: line 39: the abbreviation is used, but it is introduced later in the text - in line 44.
Response: Thank you for pointing out. PSG stands for Polysomnography. It is now introduced at line 39 in the revised version of the manuscript.
Please refer to line 39 in the revised version of the manuscript.
Comment: It is better to describe the datasets earlier in the text. I suggest doing it in line 61.
Response: Thank you for your kind suggestion. As per your suggestion, we have now shifted the dataset description to line 61 in the revised version of the manuscript.
Please refer to line 61 in the revised version of the manuscript.
Comment: line 90: it is necessary to clarify who annotated.
Response: Thank you for your kind suggestion. Certified expert sleep scorers annotated SHHS data. However, the number and names are experts have not been mentioned by the creators of the database. We have now mentioned it in the revised version of the manuscript.
Please refer to line 67 in the revised version of the manuscript.
Comment: In the first paragraph from Material, it would be good to describe the results of the right side of Table 2.
Response: Thank you so very much for your suggestion. We humbly submit that Table 2 in the previous version of the manuscript is not about results but about the sampling frequency of all channels present in the SHHS database. We have described the results in Section 4 in a detailed manner.
However, implementing your suggestion, we have removed Table 2 (sampling frequency of all channels) and have mentioned sampling frequencies relevant to used channels in a descriptive manner in the paragraph.
Please refer to line no—66 in the revised version of the manuscript.
Comment: The datasets use data on people of a certain age. The authors need to describe what limitations this places on the developed solution.
Response: Thank you for your concern. We humbly submit that no such limitation is associated with our developed solution due to age. SHHS is publicly available data, and we have used subjects of all ages and gender.
Reviewer #2
Comment: The manuscript is well written and described however author should elabore the short name when it first appears.
Response: Thank you so very much for your kind appreciation. As per your suggestion, we have elaborated on the short names in their first instance. Additionally, we have also added a list of all the acronyms and their full forms in the revised version of the manuscript.
Please refer to Table 1 in the revised version of the manuscript.
Reviewer 2 Report
The manuscript is well written and described however author should elabore the short name when it first appears.
Author Response
Comment: The manuscript is well written and described however author should elabore the short name when it first appears.
Response: Thank you so very much for your kind appreciation. As per your suggestion, we have elaborated on the short names in their first instance. Additionally, we have also added a list of all the acronyms and their full forms in the revised version of the manuscript.
Please refer to Table 1 in the revised version of the manuscript.
Reviewer 3 Report
Automatic sleep scoring is an important topic given the rise of sleep illness diagnoses, and the amount of effort required to review PSGs, as mentioned in the paper. It is also a topic that has been around since at least the early 1990s, though improved recently with the advent of machine learning algorithms.
This paper focuses on using a different approach to come up with the automated scoring, comparing it to several other published methods using large data sets for training and evaluation.
Regarding language use, the paper is quite readable, though there are some technical errors (for example, line 7 saying electrocardiogram while providing the acronym EEG, wrong formatting of the number in the Total epochs line of Table 1, etc. There is also a lot of duplication of the same content - in the abstract, introduction, methods and conclusions repeat the same content almost verbatim. This is not necessary and detracts from the message. Rather, the paper would be strengthened by explaining the underpinnings of the other approaches, and why, theoretically, this approach would be superior. This is not clear to me. It is also not clear to me whether the evaluation between the three models are similar when only comparing waking, NREM and REM sleep. How would these other models perform with this type of consolidation? Otherwise, the predictive nature seems similar or not meaningfully different.
Again, it would be useful to understand why you felt that the Tsallis feature extraction would lead to significantly better results, or the 5 level wavelet decomposition. Figure 1 could be expanded to show the difference in extraction and prediction from this method to the other ones, as well as the minimum amount of data needed for them to work as expected (with the assumption that less data is less expensive to obtain and maintain.
For the methods, was there random sampling of the cases used for training? What was that selection method? Can you provide a reference for line 130 where you state that Ensemble learning is regarding as cutting edge, and can you explain it as it pertains to your use case a little more? On line 149 I am not familiar what a confusion matrix is, can you explain? And again, with the combined NREM agreement reaching 90% for this method, we do not know what the other approaches would have been unless the analyses are made. Suggest you make them so we can have comparable results. Otherwise it looks as though this approach is not much different from the others.
Author Response
Reviewer #1
Comment: The article is devoted to a topical and important topic - the construction of an automated sleep score system. Overall, the article is well written and makes a serious impression. There are a few comments on minor corrections.
Response: Thank you so very much for your kind comments and appreciation. We have made the suggested minor corrections in the revised version of the manuscript.
Comment: line 39: the abbreviation is used, but it is introduced later in the text - in line 44.
Response: Thank you for pointing out. PSG stands for Polysomnography. It is now introduced at line 39 in the revised version of the manuscript.
Please refer to line 39 in the revised version of the manuscript.
Comment: It is better to describe the datasets earlier in the text. I suggest doing it in line 61.
Response: Thank you for your kind suggestion. As per your suggestion, we have now shifted the dataset description to line 61 in the revised version of the manuscript.
Please refer to line 61 in the revised version of the manuscript.
Comment: line 90: it is necessary to clarify who annotated.
Response: Thank you for your kind suggestion. Certified expert sleep scorers annotated SHHS data. However, the number and names are experts have not been mentioned by the creators of the database. We have now mentioned it in the revised version of the manuscript.
Please refer to line 67 in the revised version of the manuscript.
Comment: In the first paragraph from Material, it would be good to describe the results of the right side of Table 2.
Response: Thank you so very much for your suggestion. We humbly submit that Table 2 in the previous version of the manuscript is not about results but about the sampling frequency of all channels present in the SHHS database. We have described the results in Section 4 in a detailed manner.
However, implementing your suggestion, we have removed Table 2 (sampling frequency of all channels) and have mentioned sampling frequencies relevant to used channels in a descriptive manner in the paragraph.
Please refer to line no—66 in the revised version of the manuscript.
Comment: The datasets use data on people of a certain age. The authors need to describe what limitations this places on the developed solution.
Response: Thank you for your concern. We humbly submit that no such limitation is associated with our developed solution due to age. SHHS is publicly available data, and we have used subjects of all ages and gender.
Reviewer #2
Comment: The manuscript is well written and described however author should elabore the short name when it first appears.
Response: Thank you so very much for your kind appreciation. As per your suggestion, we have elaborated on the short names in their first instance. Additionally, we have also added a list of all the acronyms and their full forms in the revised version of the manuscript.
Please refer to Table 1 in the revised version of the manuscript.
Reviewer #3
Automatic sleep scoring is an important topic given the rise of sleep illness diagnoses, and the amount of effort required to review PSGs, as mentioned in the paper. It is also a topic that has been around since at least the early 1990s, though improved recently with the advent of machine learning algorithms.
This paper focuses on using a different approach to come up with automated scoring, comparing it to several other published methods using large data sets for training and evaluation.
Comment: Regarding language use, the paper is quite readable, though there are some technical errors (for example, line 7 saying electrocardiogram while providing the acronym EEG, wrong formatting of the number in the Total epochs line of Table 1, etc.
Response: Thank you so very much for your encouraging comments. We have checked critically for all these typographical errors, and they have been rectified in the revised version of the manuscript as suggested.
Comment: There is also a lot of duplication of the same content - in the abstract, introduction, methods and conclusions repeat the same content almost verbatim. This is not necessary and detracts from the message. Rather, the paper would be strengthened by explaining the underpinnings of the other approaches, and why, theoretically, this approach would be superior. This is not clear to me.
Response: Thank you for your suggestion. We have removed the duplication in the revised version of the manuscript. The method is superior because it uses both the datasets SHHS-1 and SHHS-2. To the best of our knowledge, we are the first group that have used SHHS1 and SHHS2 jointly to develop the model. Also, the method used wavelet-based Tsallis entropy features and obtained good classification performance. It is difficult to compare and may not be fair to compare our method with others because no other study has used both the combined data.
Further, the classification performance of a machine learning-based approach depends upon features extracted and the optimal classifier used. However, it cannot be predicted apriori which handcrafted features and which classifier would perform the best. It is only possible to extract and employ different features and various classifiers on a trial-and-error basis. It is very difficult for a machine learning-based method to explain theoretically why the model performed best. By the trial-and-error method, we tested many features like Shannon entropy, Tsallis entropy, Hjorth parameters (activity, mobility, complexity), log energy, norm-1, norm-2, norm-infinity, and many machine learning classifiers. After selecting EBT and tuning the hypermeter, we got the best results. As far as other approaches are concerned, our model is simpler as it uses only wavelet-based Tsallis entropy (6 features) and at the same time, the model produces comparable performances with those models that have used either SHHS-1 and SHHS2 and not both.
Comment: It is also not clear to me whether the evaluation between the three models is similar when only comparing waking, NREM, and REM sleep. How would these other models perform with this type of consolidation? Otherwise, the predictive nature seems similar or not meaningfully different.
Response: Thank you for your query. However, we apologize as we could not understand your question clearly. We failed to understand which three models you are referring to. In this study, we have developed only two models (NOT three). One is for 3-class classification (Wake vs. N vs. REM) and the other is for 5-class classification (Wake vs. N1 vs. N2 vs. N3 vs. REM). We humbly submit that the predictive nature of these models is not similar to other models.
Comment: Again, it would be useful to understand why you felt that the Tsallis feature extraction would lead to significantly better results, or the 5-level wavelet decomposition. Figure 1 could be expanded to show the difference in extraction and prediction from this method to the other ones, as well as the minimum amount of data needed for them to work as expected (with the assumption that less data is less expensive to obtain and maintain.
Response: Thank you for your suggestion. To develop the model, we have tested many features like Shannon entropy, Tsallis entropy, Hjorth parameters (activity, mobility, complexity), log energy, norm-1, norm-2, norm-infinity, and non-linear features using a trial-and-error approach. Eventually, we observed that the Tsallis entropy feature led to significantly better results. Similarly, for wavelet decomposition, we experimented with decomposition from level-2 to level-7 and observed that 5-level wavelet decomposition was optimally performing.
We would like to submit that Figure 1 humbly describes the flow of the proposed study optimally and exhaustively, and hence it seems a bit tricky to expand it further. We have tried our best to show all the steps involved in our methodology. We have extracted the wavelet-based Tsallis feature from EEG, EMG, and EOG signals employing the SHHS database and used the Ensemble Bagged Tree classifier with 10% holdout validation for prediction. This study has taken full-night recordings of 8326 subjects to get the expected results. We agree with your comment that less data is less expensive to obtain and maintain. Nevertheless, training the model with a vast database is always advisable in a supervised machine learning-based model because diverse and large databases may have different patterns which may not be present in one (small) set but present in another subset. Therefore, the gold standard for training purposes is more data, better the learning. Further, it seems complicated to comment on the minimum data required for other studies to get the expected results.
Comment: For the methods, was there random sampling of the cases used for training? What was that selection method?
Response: Thank you for your query. In machine learning, 10% holdout validation is a standard approach to test the performance of the model [20,23]. We have selected 10% subjects for testing and 90% subjects for testing randomly. Further, it is to be noted that 10% subjects involved in testing and 90% subjects involved in training are mutually exclusive, i.e., the subjects which are used in testing are not used for training and vice versa. Therefore, the model developed is free from any bias.
Comment: Can you provide a reference for line 130 where you state that Ensemble learning is regarding as cutting edge, and can you explain it as it pertains to your use case a little more?
Response: Thank you for your suggestion. As per your suggestion, we have added references [23, 24] on line 129 and expanded more on ensemble learning in the revised version of the manuscript.
Please refer to lines 127 – 132 in the revised version of the manuscript.
“A bagging method and decision tree classifier are combined in the EBT classifier. Bagging is a strategy that uses bootstrap sampling to decrease decision tree variance and increase learning algorithm efficiency by producing a group of learning algorithms that are trained in succession. It employs arbitrary sampling with a replacement rather than a conventional averaging of all outcomes from multiple decision trees.”
Comment: On line 149 I am not familiar what a confusion matrix is, can you explain?
Response: Thank you for your query. The confusion matrix is a widely used metric when it comes to classification problems. The confusion matrix for the 3-class sleep stage classification for both datasets is shown in Table 7. It suggests that for the SHHS-1 dataset, 92% of actual wake stage epochs were correctly predicted as wake stage, 7% of wake stage epochs were incorrectly predicted as N, and 1% wake stage epochs were incorrectly predicted as REM stage epochs.
In the revised version of the manuscript, we have added a reference [26, 27] besides the Confusion matrix for better readability and understanding.
Comment: with the combined NREM agreement reaching 90% for this method, we do not know what the other approaches would have been unless the analyses are made. Suggest you make them so we can have comparable results. Otherwise it looks as though this approach is not much different from the others.
Response: Thank you for your suggestion. In this study, we have used both the SHHS-1 and SHHS-2 datasets and have done 3-class as well as 5-class classification. Other studies have used either SHHS-1 or SHHS-2 subsets. Also, none of them have done 3-class classification, unlike us. So, it appears difficult for us to compare our results with them as they have not done 3-class classification.
We politely submit that this method is different in terms of feature extracted, machine learning algorithm used and it also produces good results. Kappa values above 0.65 are considered to be an indicator of good agreement. The kappa value is above 0.77 for the model developed by us. Hence the proposed method has produced an acceptable model. This method is entirely different from others in terms of features used. No other methods have been used:
(1) both SHHS-1 and SHHS-2 datasets combined;
(2) we have done 3-class classification and 5-class classification, both using the SHHS dataset;
(3) We have used wavelet-based Tsallis features with the EBT classifier.